# An Injecting Molding Method for Forming the HFRP/PA6 Composite Parts

**DOI:** 10.3390/polym14235085

**Published:** 2022-11-23

**Authors:** Bin Xu, Meng-Yang Wei, Xiao-Yu Wu, Jian-Guo Lei, Zhi-Wen Zhou, Lian-Yu Fu, Li-Kuan Zhu

**Affiliations:** 1Shenzhen Key Laboratory of High Performance Nontraditional Manufacturing, Shenzhen University, 3688 Nanhai Avenue, Nanshan District, Shenzhen 518060, China; 2Shenzhen Jinzhou Precision Technology Corp., Shenzhen 518060, China

**Keywords:** HFRP/PA6 composite part, injection molding, connecting strength

## Abstract

Carbon/glass fiber-reinforced polymer hybrid composite (HFRP) has the advantages of a light weight and high strength. For the lightweight design of automobile parts, composite parts made of HFRP and polymer materials are increasingly in demand. The method of the injection molding is usually adopted to fabricate composite part with HFRP and polymer materials. The connecting strength between the two materials has an important influence on the service life of the composite part. In this paper, HFRP and polyamide-6 (PA6) were used to fabricate a composite part by the injection molding method. In order to improve the connecting strength between HFRP and PA6, a kind of micro-grooves was fabricated on the HFRP surface. The micro-grooves on the surface of the HFRP provided sufficient adhesion and infiltrating space of molten PA6 material into the mold. In addition, the glass fiber in HFRP can also be used as nucleating agent to facilitate the rapid crystallization of PA6. The micro-grooves on the surface of HFRP were embedded into PA6 like nails, which could improve the connecting strength at the interface effectively. The paper investigated the effects of mold temperature, injection pressure, holding pressure and holding time on the injection quality and connecting strength of composite parts in detail. With a mold temperature of 240 °C, an injection pressure of 8 MPa, a holding pressure of 8 MPa and a holding time of 3 s, the maximum tensile strength of 10.68 MPa was obtained for the composite part. At the effect of micro-grooves, the tensile strength of the composite part could be increased by 126.27%.

## 1. Introduction

Carbon fiber-reinforced plastics (CFRP) have been widely used in many fields such as wind energy, aerospace and automobiles due to their excellent mechanical, thermal and electrical properties [1,2]. As the non-metallic material with the greatest application potential in modern industry, carbon fiber-reinforced plastics (CFRP) have excellent corrosion resistance and fatigue resistance [3,4]. In addition, carbon/glass fiber-reinforced polymer hybrid composite (HFRP) has also been deeply studied due to the high mechanical properties and corrosion resistance properties of carbon fiber and the low price and high deformation properties of glass fiber [5]. *Xian* et al. realized a synergetic bearing mechanism between carbon and glass fibers through the random fiber hybrid technology. The incongruous stress concentration of the carbon/glass fiber/resin interface was largely relieved, which generated a higher tensile and flexural strength [6]. *Barjasteh* et al. investigated the effect of moisture absorption on the mechanical and thermal properties of hybrid composite rods designed to support overhead transmission lines. Sorption curves were obtained for both hybrid and non-hybrid composite rods to determine characteristic parameters [7]. In practical applications, CFRP often needs to be joined with other materials to form functionally enhanced components [8,9]. Polyamide-6 (PA6) is a kind of nylon material with good wear resistance and self-lubricating properties, which is widely used in automobile parts, electronic equipment and packaging materials [10]. For the excellent properties of HFRP and PA6, the composite parts made of these two materials have been widely used in the lightweight design of automobiles [11].

By using the adhesive bonding method, Ribeiro et al. adopted adhesive XNR6852 to join CFRP with aluminum alloy, and the tensile strength of the joints could reach 21 MPa [12]. Guo et al. proposed an ultrasonic additive manufacturing method to join CFRP and aluminum, with the tensile strength of joints reaching 129.5 MPa [13]. Using the laser welding method, Arkhurst et al. fabricated CFRP-AZ31 Mg alloy joints, and they thoroughly studied the effect of thermal oxidation on the joining strength [14]. By the friction stir interlocking method, Wang et al. joined AZ31 magnesium alloy and CFRP sheets with a thermoplastic matrix [15], and the joints could sustain the shear tensile from 80 to 100 N/mm. With the hybrid bonded-riveted method, Li et al. fabricated CFRP/Al joints and studied the effects of high-speed loading and environmental factors on the dynamic responses of the joints [16]. Can et al. analyzed the dynamic behavior of the screwed joints in CFRP composite laminates, and the experimental results showed that the dynamic failure loads for the screwed joints ranged from 27.55 to 78.12 kN [17]. Cococcetta et al. studied the effect of the post-processing technique on 3D-printed CFRP composites, and they found that the cryogenic machining method could bring down the surface roughness from 10 μm to 1 μm [18]. Tian et al. conducted Extensive UW tests on 2.3 mm-thick nylon6 composites with 30 wt% carbon fiber to investigate the effect of the preload on weld quality and found that the strength was increased by 18.7% for the normal joint with a preload of 200 N made with optimal welding parameters [19]. A lot of research works have been conducted on the properties of the HFRP composite parts; the results showed that the mechanical properties and service life of those composite parts are limited due to the operating conditions in those forming technologies. Hence, in order to guarantee the properties of HFRP composite parts, we proposed a new forming method for manufacturing the composite parts based on the HFRP and Polyamide-6 (PA6).

On the other hand, the forming method of PA6 includes the injection molding, hot-pressing molding and injection compression molding [20,21]. Injection molding is a simple process with a low production cost and a high molding quality which has been widely applied in the molding of polymer parts [22,23]. In this paper, composite parts were made of HFRP and PA6 using the injection molding technique. As shown in Figure 1a, HFRP/PA6 composite parts were obtained by injection molding using the HFRP without micro-grooves. Based on previous experimental data, the maximum tensile strength of the composite part could reach 4.72 MPa (Figure 1b). The connecting strength at the interface of HFRP and PA6 had an important influence on the mechanical properties and service life of composite parts. In order to further improve the connecting strength of HFRP and PA6, micro-grooves were fabricated on the surface of HFRP by the hot-pressing method, and then it was applied in the injection molding of composite parts.

## 2. Materials, Experimental Equipment and Processing Technology

### 2.1. Materials and Experimental Equipment

The physical properties of PA6 are shown in Table 1. The HFRP used in this paper was provided by Shenzhen Silver Basis Technology Co., Ltd. (Shenzhen, China). It was composed of carbon fiber (CF), glass fiber (GF), epoxy resin and a small amount of PA6, and its physical properties are shown in Table 2.

The hot press machine (SZU1.0, Shenzen University, Shenzhen, China) was used to fabricate micro-groove array on the surface of HFRP. The micro-grooves had a bottom diameter of 94 μm, a depth of 180 μm and an inclination angle of 12°. The injection cavity was machined on the mold core by the 5-axis CNC machining center (DMU40, DEMAGE, Bavaria, Germany). The composite parts with PA6 and HFRP were formed by the injection molding machine (Babyplast 6/10P, Cronoplast Sl, Barcelona, Spain). The cross-sectional profile and fracture morphology of composite parts were observed using the stereo microscope (VHX-2000, KEYENCE, Yokohama, Japan). The universal tensile machine (Z050TEW, ZWICK, Ulm, Germany) was used to carry out the tensile tests of composite parts with a preload force of 5 N and a tensile speed of 0.01 mm/s.

### 2.2. Steps of the Experiment

The process flow of the composite part is shown in Figure 2, and the specific process is described as follows:(1)The injection cavity was machined on the mold core by the 5-axis CNC machining center. In order to facilitate the subsequent tensile test of composite parts, the injection cavity with an H shape was designed (Figure 2a). The injection cavity had a length of 10 mm, a depth of 2 mm and a draft angle of 3°.(2)The HFRP with a micro-groove array and gaskets were placed into the injection cavity based on a certain order (Figure 2b). The operating parameters for the injection molding process were set, and then the injection molding was performed. During the injection molding process, molten PA6 entered the injection cavity through the pouring gate and solidified on the HFRP surface quickly (Figure 2c).(3)After pressure holding and cooling, the injection molded composite part was obtained. The micro-grooves on the HFRP surface were embedded into PA6 like nails (Figure 2d), which could effectively improve the connecting strength between HFRP and PA6 in the composite part.(4)The tensile strength of the composite parts was measured using a universal tensile machine. The tensile force was increased from a preload of 5N and at a tensile speed of 0.01 m/s until the joint surface of the composite parts was separated. In order to obtain more reliable experimental results, each tensile curve was obtained from the average data of three same samples.

**Figure 2 polymers-14-05085-f002:**
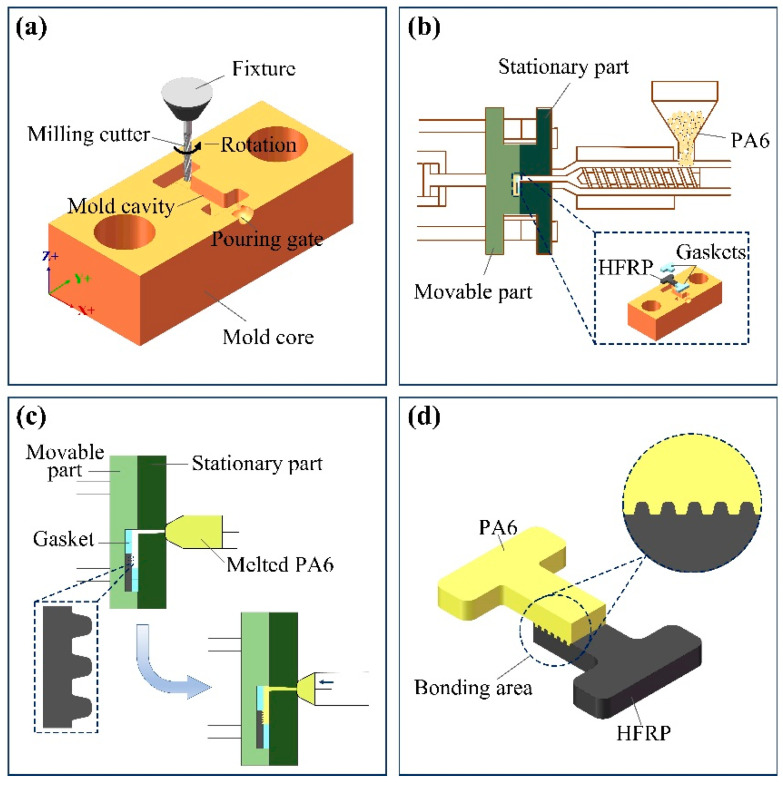
Process flow of the composite part. (**a**) fabrication of the mold core by the 5-axis CNC machining center; (**b**) placement of the HFRP and gaskets; (**c**) injection molding; (**d**) composite part.

The operating parameters of the injection molding process include the mold temperature, injection pressure, holding pressure and holding time, which have an important influence on the molding quality and connecting strength of the composite part. Based on the single factor experiment method, the operating parameters with 18 different groups (Table 3) were considered to study the effects of the mold temperature, injection pressure, holding pressure and holding time on the molding quality and connecting strength of the composite part. The length of the bonding zone of the composite parts was 2 mm, its width was 2 m and its height was 2 mm. The overall length of the composite parts was 10 mm.

## 3. Results and Discussion

### 3.1. Effect of the Mold Temperature

At the effect of the mold temperature, PA6 changed from a solid state to a melting state, presenting enough flowability. Therefore, the mold temperature played an important role in providing PA6 with sufficient flowability. In order to study the effect of the mold temperature on the molding quality of the composite part, the composite parts with HFRP and PA6 were formed by the injection molding method using different mold temperatures of 230 °C, 235 °C, 240 °C, 245 °C and 250 °C. During the experiment, the injection pressure was set to 8 MPa, the holding pressure was set to 5 MPa and the holding time was set to 2.5 s. The tensile strength of the composite part was tested by the universal tensile machine. The cross-sectional profile and fracture morphology of the composite part were determined by the stereo microscope. The experimental results are shown in Figure 3.

As shown in Figure 3, the micro-groove structure on the HFRP surface had been completely filled with PA6, which could guarantee the injection molding quality of the composite part. When the mold temperature increased from 230 °C to 240 °C, the tensile strength of the composite part also increased from 5.59 MPa to 8.54 MPa. By increasing the mold temperature, the residual CF and GF on the fracture surface of the composite part also increased. When the mold temperature was lower than 240 °C, increasing the mold temperature improved the flowability of molten PA6 in the mold core cavity. As a result, the permeating degree and mutual infiltration degree of PA6 and HFRP were enhanced, which could improve the tensile strength of the composite part. When the mold temperature increased from 240 °C to 250 °C, the tensile strength of the composite part decreased from 8.54 MPa to 7.27 MPa. When the mold temperature was higher than 240 °C, increasing the mold temperature probably caused the fracture of hydrogen bonds in PA6, leading to the degradation of the connecting strength of PA6. Therefore, an excessively high mold temperature led to the degradation of the tensile strength of the composite part. In addition, an excessively high mold temperature will also cause the formation of bubbles at the junction section between the HFRP and PA6, leading to the degradation of the connecting strength of the composite part. Therefore, in order to ensure the connecting strength of the composite part, a mold temperature of 240 °C was proposed based on the results above.

### 3.2. Effect of the Injection Pressure

At the effect of the injection pressure, the molten PA6 entered the mold core cavity through the pouring gate. In order to investigate the effect of the injection pressure on the molding quality of composite parts, the injection molding experiments were carried out on the HFRP and PA6 at different injection pressures. The injection pressures of 6 MPa, 7 MPa, 8 MPa, 9 MPa and 10 MPa were investigated separately. In the experiments, the mold temperature was 240 °C, the holding pressure was 5 MPa and the holding time was 2.5 s. The experimental results are shown in Figure 4.

When the injection pressure was 6 MPa, gaps were observed at the interface of HFRP and PA6. Correspondingly, the micro-grooves on the surface of HFRP were not completely filled with PA6, which caused the poor injection molding quality of the composite part. As the injection pressure increased, the PA6 could fill the micro-groove structure on the HFRP surface more easily, and the injection molding quality of the composite part could be increased efficiently. When the injection pressure increased from 6 MPa to 8 MPa, the residual CF and GF on the fracture surface of the composite part increased gradually, and the tensile strength increased from 5.45 MPa to 8.54 MPa. However, as the injection pressure increased over 8 MPa, the residual CF and GF on the fracture surface of composite parts started to decrease gradually, and the tensile strength also decreased. When the injection pressure was 10 MPa, flying edges occurred in the composite parts, which would decrease the injection molding quality.

The experimental results indicated that too high or too low of an injection pressure was not favorable for improving the injection molding quality and tensile strength of composite parts. When the injection pressure was too low, the pressure in the mold core cavity was insufficient, leading to the insufficient filling of PA6 and thus resulting in the gaps at the interface of the HFRP and PA6 connection. When the injection pressure was suitable, the arrangement between PA6 molecules and glass fibers was more compact, thus facilitating the formation of the strong condensed structure at the interface of HFRP and PA6. Moreover, a proper injection pressure could promote the heterogeneous nucleation between the glass fiber and PA6 molecules, which was beneficial to improving the tensile strength of the composite part.

The relation between the mold filling rate (*Q*) and the injection pressure (*P*) during injection molding was given by:(1)$$Q=\frac{P}{\eta K}$$
where η is polymer viscosity and *K* is mold resistance. According to Equation (1), the injection pressure (*P*) is proportional to the mold filling rate (*Q*). With the increase in the injection pressure, the mold filling rate of the molten PA6 entering the core cavity increases. With the increase in the mold filling rate, the contact time between the molten PA6 and the cold cavity wall shortens, thus limiting the occurrence of the freezing phenomena and the short shot phenomena. At this condition, the molten PA6 can fill the injection molding cavity more fully so as to effectively guarantee the injection molding quality and tensile strength of the composite part. However, as the injection pressure (*P*) is too high, the mold filling rate (*Q*) reaches an excessive value. At these kinds of conditions, the molten PA6 is prone to cause a wall slip phenomenon in the injection mold cavity, which decreases the injection molding quality and tensile strength of the composite part.

Therefore, in order to ensure the injection molding quality of the composite part, the injection pressure should be kept within an appropriate range. By comparing the experimental data, an injection pressure of 8 MPa was proposed in the paper.

### 3.3. Effect of the Holding Pressure

At the final stage of the injection molding process, it is necessary to apply certain pressure on the PA6 material in the molding core cavity to guarantee the molding quality of the composite part. This pressure was named by the holding pressure. In order to analyze the effect of the holding pressure on the molding quality of the composite part, the injection molding process was performed on the HFRP and PA6 at different holding pressures. The holding pressure was set to 4 MPa, 5 MPa, 6 MPa, 7 MPa and 8 MPa, respectively. During the experiments, the mold temperature was set to 240 °C, the injection pressure was set to 8 MPa and the holding time was set to 2.5 s. The experimental results are shown in Figure 5.

According to the results shown in Figure 5, the tensile strength of the composite part tended to increase by increasing the holding pressure. When the holding pressure was 4 MPa, the smallest value of the tensile strength for the composite part was reached at 6.43 MPa. When the holding pressure was 6 MPa, the tensile strength for the composite part was reached at 8.29 MPa. As the holding pressure continued to increase, a part of micro-grooves on the surface of HFRP will break, resulting in a small decrease in the tensile strength. However, as the holding pressure increased further, the previously broken part recombined, which caused the tensile strength to continue to increase. When the holding pressure was 8 MPa, the largest value of the tensile strength for the composite part was about 9.67 MPa. At the effect of the holding pressure, the molecules of the molten PA6 began to crystallize, and the molecular chains aligned with each other and finally formed the composite part with the HFRP. When the holding pressure was low, the compression of the molten PA6 was insufficient, resulting in the low density of the solidified PA6. Under this condition, the composite part had a low injection molding quality and an insufficient tensile strength. When the holding pressure continued to increase within the proper range, the injection molding quality of the composite part increased, and this was the reason why the tensile strength presented an upward trend. When the holding pressure was 8 MPa, there were a lot of CF and GF remaining in the fracture surface of the composite part. Therefore, the tensile strength between PA6 and HFRP was improved, effectively ensuring the mechanical properties of composite parts.

### 3.4. Effect of the Holding Time

In order to ensure the injection molding quality of the composite part, the holding pressure needs to be maintained for a certain time, and this time is named by the holding time. A proper holding time is necessary to avoid the adverse effect of PA6 volume shrinkage caused by the injection molding process. To study the effect of the holding time on the injection molding quality of the composite part, the injection molding processes of the HFRP and PA6 were carried out at different holding times. The holding time was 2 s, 2.5 s, 3 s, 3.5 s and 4 s, respectively. During the experiment, the mold temperature was set to 240 °C, the injection pressure was set to 8 MPa and the holding pressure was set to 8 MPa. The experimental results are shown in Figure 6.

According to the experimental results shown in Figure 6, when the holding time increased from 2 s to 3 s, the injection molding quality of the composite part increased. Under this condition, the tensile strength of the composite part increased from 6.55 MPa to 10.68 MPa. When the holding time was 3 s, there were a lot of CF and GF in the fracture surface of the composite part. However, as the holding time continued to increase from 3 s to 4 s, the tensile strength of the composite parts decreased from 10.68 MPa to 5.84 MPa. Therefore, the hold time should be kept in a proper range to promote the increase in the molding quality and tensile strength of the composite part.

At the pressure holding stage, the molten PA6 crystallized again and solidified gradually. When the holding time was short, the PA6 was heated unevenly in different areas, thus forming holes inside the PA6 easily and resulting in the decrease in the injection molding quality and tensile strength. As the holding time increased, PA6 crystallized more fully and solidified on the HFRP surface. Therefore, when the holding time was 3 s, the tensile strength of the composite part could reach 10.68 MPa. However, when the pressure holding time was too long, the pressure in the cavity of PA6 increased, thus easily resulting in the uneven distribution of stress inside the composite part, which could cause defects such as flying edges and limit the mechanical properties and tensile strength of the composite part.

## 4. Conclusions

For the lightweight design of automobile parts, composite parts made of HFRP and PA6 are becoming increasingly common. To improve the connecting strength between HFRP and PA6, the paper applied HFRP with micro-grooves in the injection molding process. The main conclusions of this study are as follows:(1)The micro-groove structures effectively increased the injection molding area of the composite parts, which could increase the connection strength between HFRP and PA6. In order to increase the molding properties of the micro-groove structures, the hot-pressing direction should be perpendicular to the fiber direction, which could guarantee the direction of the micro-grooves parallel to the fiber direction.(2)A mold temperature of 240 °C, an injection pressure of 8 MPa, a holding pressure of 8 MPa and a holding time of 3 s were proven to be suitable for increasing the injection molding quality of the composite part. At this condition, PA6 could completely fill into the micro-grooves on the HFRP surface.(3)At the effect of the micro-groove structure on the HFRP surface, the tensile strength of the composite part could be increased from 4.72 MPa to 10.68 MPa, and the tensile strength was increased by 126.27%.

## Figures and Tables

**Figure 1 polymers-14-05085-f001:**
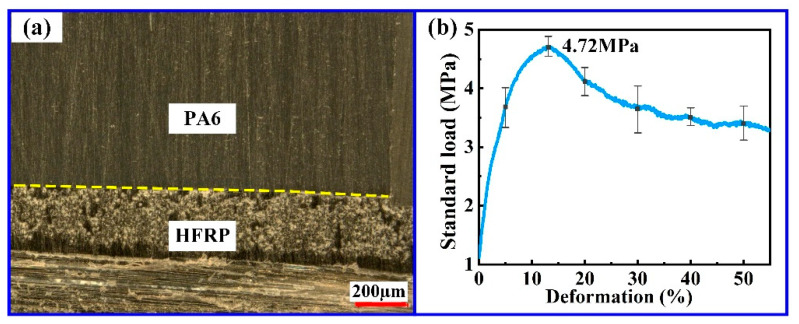
Composite parts: (**a**) cross-sectional profile; (**b**) tensile curve.

**Figure 3 polymers-14-05085-f003:**
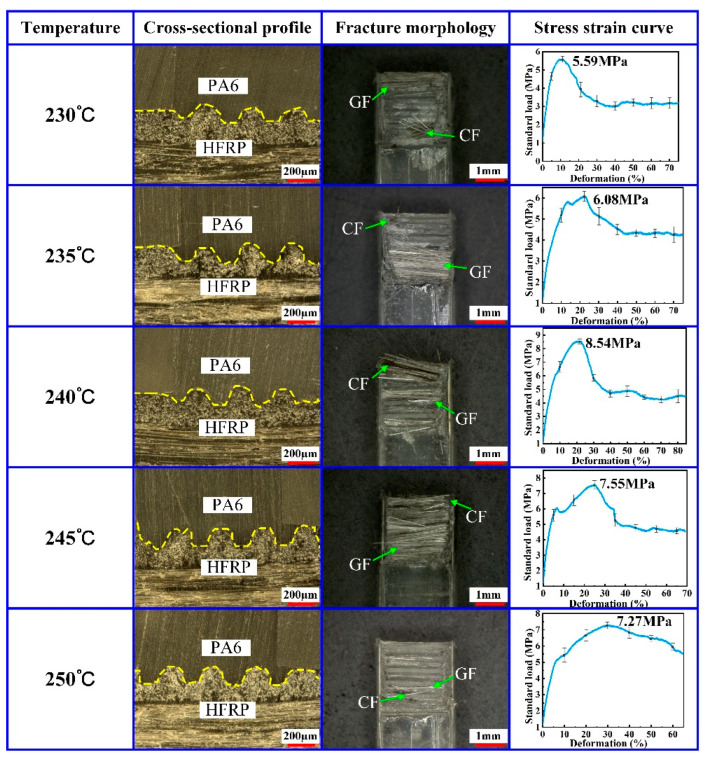
Effect of mold temperature on the injection molding quality of composite parts.

**Figure 4 polymers-14-05085-f004:**
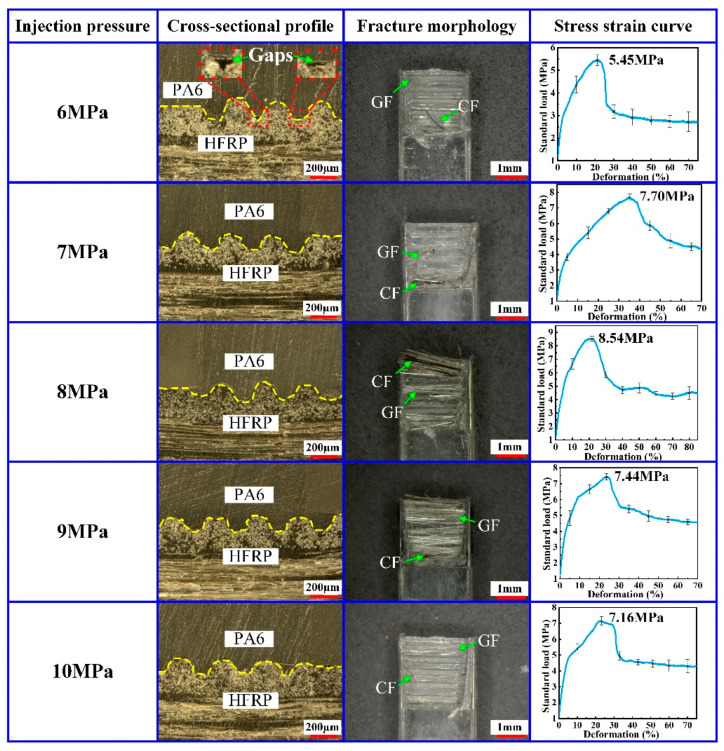
Effect of injection pressure on the injection molding quality of composite parts.

**Figure 5 polymers-14-05085-f005:**
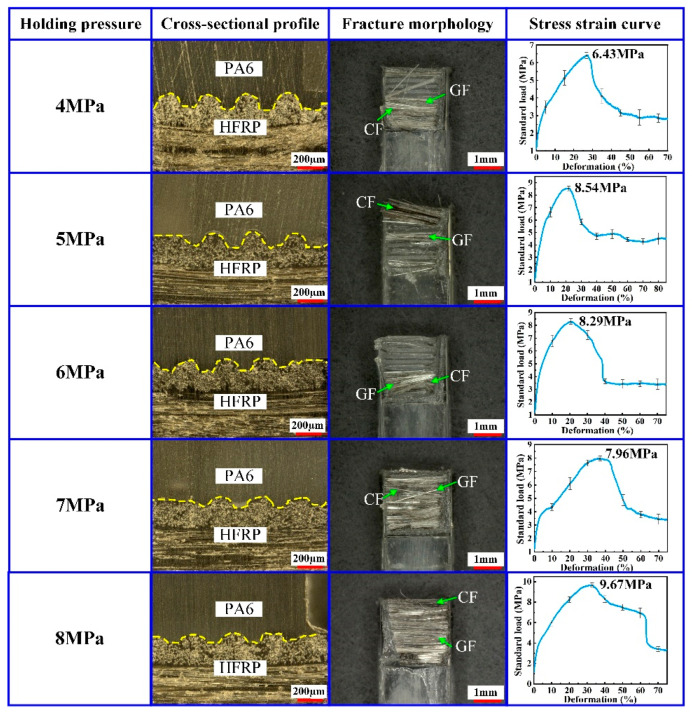
Effect of the holding pressure on the injection molding quality of composite parts.

**Figure 6 polymers-14-05085-f006:**
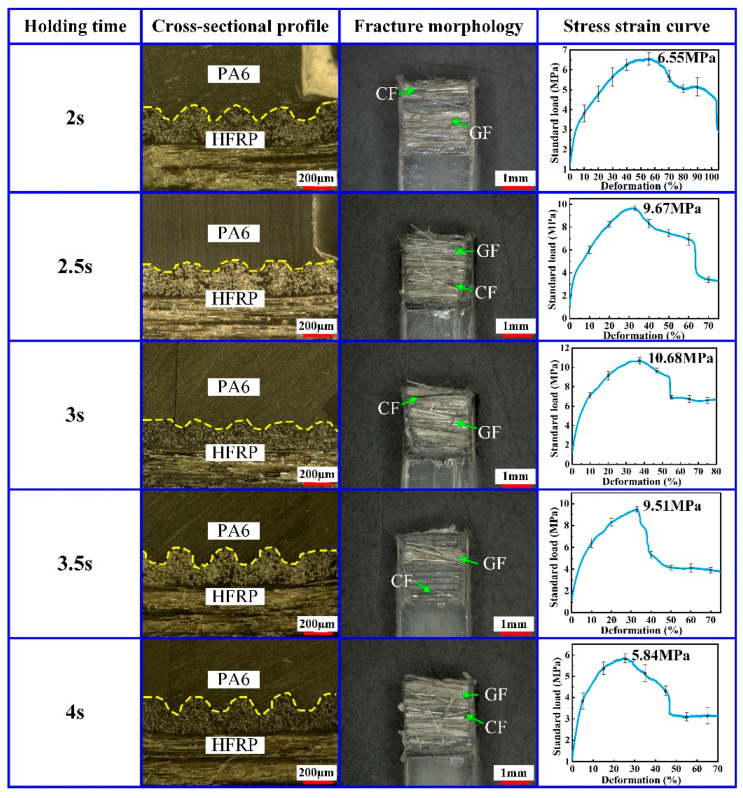
Effect of holding time on the injection molding quality of the composite part.

**Table 1 polymers-14-05085-t001:** The physical properties of the PA6.

Density (g/cm^3^)	Melting Point (°C)	Bending Strength (MPa)	Water Absorption (%)
1.13	215	90.0	3.5

**Table 2 polymers-14-05085-t002:** The physical properties of the HFRP.

Tensile Strength (GPa)	Tensile Modulus (GPa)	Content of Fiber (%)	Content of PA6 (%)
4.3	230	67.14	1~5%

**Table 3 polymers-14-05085-t003:** Operating parameters for the injection molding process.

No.	Mold Temperature	Injection Pressure	Holding Pressure	Holding Time
	*T* (°C)	$F_{1}\;(\mathrm{Mpa})$	$F_{2\;}(\mathrm{Mpa})$	*t* (s)
1	230	8	5	2.5
2	235	8	5	2.5
3	240	8	5	2.5
4	245	8	5	2.5
5	250	8	5	2.5
6	240	6	5	2.5
7	240	7	5	2.5
8	240	9	5	2.5
9	240	10	5	2.5
10	240	8	3	2.5
11	240	8	4	2.5
12	240	8	6	2.5
13	240	8	7	2.5
14	240	8	8	2.5
15	240	8	8	2
16	240	8	8	3
17	240	8	8	3.5
18	240	8	8	4

## Data Availability

The data presented in this study are available on request from the corresponding author.

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
