# Peer review of "An Injecting Molding Method for Forming the HFRP/PA6 Composite Parts"

_polymers, 2022, doi:10.3390/polym14235085_

Round 1

Reviewer 1 Report

A kind of micro-grooves were fabricated on the CFRP surface to improve the connecting strength between CFRP and PA6. Different process parameters are adopted to analyze their effects on interface properties. The following major comments should be responded to improve the quality of the paper. 

1. Abstract, the tensile strength of composite part was improved through a kind of micro-grooves. Please provide the improvement mechanism analysis for tensile strength. In addition, will the micro-grooves on the surface of CFRP produce initial defects and stress concentration? How to consider such a problem?

2. Introduction, in addition to excellent mechanical properties, thermal properties and electrical properties in the first paragraph, the corrosion resistance and fatigue resistance of CFRP are also the critical factors to be considered in many application fields compared with other metal materials. This is because CFRP inevitably encounters the long-term effects of environment and cyclic loading during its service. Therefore, fatigue and corrosion resistances are important index to guarantee the long-term safe use of CFRP. Please review some recent research work on fatigue and corrosion resistances for CFRP, such as International Journal of Fatigue, 2020, 134: 105480. Composite Structures, 2021, 256: 113058. Composite Structures, 2022, 293, 115719.

3. On the summary of interface bonding between CFRP and other metal materials, it is suggested to provide some relevant mechanism analysis on interface bonding and the improvement methods. In addition, the analysis on molding process of PA6 should be enriched.

4. Carbon fiber and glass fiber hybrid reinforced epoxy resin composites are adopted in this paper. However, there is no relevant summary work on hybrid composite in the introduction. It is suggested that the authors add this summary related to this paper (such as R. Guo and GJ Xian    E. Barjasteh and S.R. Nutt). In addition, the innovation of this research work should be further highlighted.

5. In part 2.1, “The CFRP used in this paper was composed of carbon fiber (CF), glass fiber (GF), epoxy resin…”, there is some glass fiber in CFRP. It should not be called CFRP. Please adjust this statement. In addition, the overall volume fraction of fiber is 67%. What are the volume fractions of carbon fiber and glass fiber?

6. In table 2, CFRP contains 1% ~ 5% PA6. In general, PA6 is a solid state at room temperature, so what is its existing form in CFRP?

7. The micro-grooves had a bottom diameter of 94 μm, a depth of 180 μm and an inclination angle of 12°. Will the CFRP surface be damaged during the preparation of the micro-grooves, such as resin cracking, fiber breaking and interface debonding?

8. The tensile strength of composite part is mentioned in the abstract. However, the tensile test method of composite parts is not given in the part 2. Please make relevant supplement. In addition, Table 3 should be placed in part 2.

9. How many samples are adopted to obtain the stress-strain curves in Fig. 3? Generally, the average value of at least three samples shall be taken.

10. In the last part of the results and discussion, it is suggested that the authors summarize the optimal process parameters to obtain the maximum interface performance.

11. The conclusion part should be further enriched according to the content of this paper, including 3-4 small conclusions.

Author Response

Comment 1: Abstract, the tensile strength of composite part was improved through a kind of micro-grooves. Please provide the improvement mechanism analysis for tensile strength. In addition, will the micro-grooves on the surface of CFRP produce initial defects and stress concentration? How to consider such a problem?

Response:

The micro-grooves on the surface of HFRP could provide sufficient adhesion and infiltrating space for molten PA6 material into the mold. In addition, the glass fiber in HFRP can act as a kind of nucleating agent to facilitate the rapid crystallization of PA6. Those words were also added into the Abstract Part in the revised manuscript.

When the micro-grooves were prepared on the surface of HFRP, it did cause initial defects and stress concentration on the surface. In order to reduce this kind of effect, we prepared the micro-grooves along the fiber direction with the hot-pressing mold.

Comment 2: Introduction, in addition to excellent mechanical properties, thermal properties and electrical properties in the first paragraph, the corrosion resistance and fatigue resistance of CFRP are also the critical factors to be considered in many application fields compared with other metal materials. This is because CFRP inevitably encounters the long-term effects of environment and cyclic loading during its service. Therefore, fatigue and corrosion resistances are important index to guarantee the long-term safe use of CFRP. Please review some recent research work on fatigue and corrosion resistances for CFRP, such as International Journal of Fatigue, 2020, 134: 105480. Composite Structures, 2021, 256: 113058. Composite Structures, 2022, 293, 115719.

Response:

More words and references were cited in the Introduction part : “As the non-metallic material with the greatest application potential in modern industry, carbon fiber reinforced plastics (CFRP) have excellent corrosion resistance and fatigue resistance [3,4]. In addition, carbon / glass fiber reinforced polymer hybrid composite (HFRP) has also been deeply studied due to both the high mechanical properties and corrosion resistance properties of carbon fiber, and the low price and high deformation properties of glass fiber [5], Xian et al. realized synergetic bearing mechanism between the carbon and glass fibers through the random fiber hybrid technology. Incongruous stress concentration of carbon/glass fiber/resin interface was largely relieved, which generated higher tensile and flexural strength [6]. Barjasteh et al. investigated the effect of moisture absorption on the mechanical and thermal properties of hybrid composite rods designed to support overhead transmission lines. Sorption curves were obtained for both hybrid and non-hybrid composite rods to determine characteristic parameters [7].”

  1. Ding, J.L.; Cheng, L.; Chen, X.; Chen, G.; Liu, K. A review on ultra-high cycle fatigue of CFRP. Composite Structures 2021, 256, 113058. https://doi.org/10.1016/j.compstruct.2020.113058.
  2. Xian, G.J.; Guo, R.; Li, C.G.; Wang, Y.J.; Mechanical performance evolution and life prediction of prestressed CFRP plate exposed to hygrothermal and freeze-thaw environments. Composite Structures 2022, 293, 115719. https://doi.org/10.1016/j.compstruct.2022.115719.
  3. Li, C.G.; Yin, X.L.; Liu, Y.C.; Guo, R.; Xian, G.J. Long-term service evaluation of a pultruded carbon/glass hybrid rod exposed to elevated temperature, hydraulic pressure and fatigue load coupling. International Journal of Fatigue 2020, 134, 105480. https://doi.org/10.1016/j.ijfatigue.2020.105480.
  1. Xian, G.J.; Guo, R.; Li, C.G. Combined effects of sustained bending loading, water immersion and fiber hybrid mode on the mechanical properties of carbon/glass fiber reinforced polymer composite. Composite Structures 2022, 281, 115060. https://doi.org/10.1016/j.compstruct.2021.115060.
  2. Barjasteh, E.; Nutt, S.R.; Moisture absorption of unidirectional hybrid composites. Composites: Part A 2012, 43, 158-164. https://doi.org/10.1016/j.compositesa.2011.10.003.

Comment 3: On the summary of interface bonding between CFRP and other metal materials, it is suggested to provide some relevant mechanism analysis on interface bonding and the improvement methods. In addition, the analysis on molding process of PA6 should be enriched.

Response:

More words and reference were added into the Introduction to those issues pointed out by the reviewer:“Tian et.al conducted an Extensive UW tests on 2.3-mm-thick nylon6 composites with 30 wt% carbon fiber to investigate the effect of the preload on weld quality, and found that the strength was increased by 18.7% for the normal joint with preload of 200 N made with optimal welding parameters [19]. Lots of researching works have been done on the properties of the HFRP composite parts, the results showed that the mechanical properties and service life of those composite parts are limited due to the operating conditions in those forming technologies. Hence, in order to guarantee the properties of HFRP composite parts, we proposed a new forming method for manufacturing the composite parts based on the HFRP and Polyamide-6 (PA6).”

  1. Tian, Z.; Zhi, Q.; Feng, X.; Zhang, G.; Li, Y.; Liu, Z. Effect of Preload on the Weld Quality of Ultrasonic Welded Carbon-Fiber-Reinforced Nylon 6 Composite. Polymers 2022, 14, 2650. https://doi.org/10.3390/polym14132650.

Comment 4: Carbon fiber and glass fiber hybrid reinforced epoxy resin composites are adopted in this paper. However, there is no relevant summary work on hybrid composite in the introduction. It is suggested that the authors add this summary related to this paper (such as R. Guo and GJ Xian    E. Barjasteh and S.R. Nutt). In addition, the innovation of this research work should be further highlighted.

Response:

More relevant works on the hybrid composite properties were added into the Introduction: “In addition, carbon / glass fiber reinforced polymer hybrid composite (HFRP) has also been deeply studied due to both the high mechanical properties and corrosion resistance properties of carbon fiber, and the low price and high deformation properties of glass fiber [5], Xian et al. realized synergetic bearing mechanism between the carbon and glass fibers through the random fiber hybrid technology. Incongruous stress concentration of carbon/glass fiber/resin interface was largely relieved, which generated higher tensile and flexural strength [6]. Barjasteh et al. investigated the effect of moisture absorption on the mechanical and thermal properties of hybrid composite rods designed to support overhead transmission lines. Sorption curves were obtained for both hybrid and non-hybrid composite rods to determine characteristic parameters [7].”

5. Li, C.G.; Yin, X.L.; Liu, Y.C.; Guo, R.; Xian, G.J. Long-term service evaluation of a pultruded carbon/glass hybrid rod exposed to elevated temperature, hydraulic pressure and fatigue load coupling. International Journal of Fatigue 2020, 134, 105480. https://doi.org/10.1016/j.ijfatigue.2020.105480.

  1. Xian, G.J.; Guo, R.; Li, C.G. Combined effects of sustained bending loading, water immersion and fiber hybrid mode on the mechanical properties of carbon/glass fiber reinforced polymer composite. Composite Structures 2022, 281, 115060. https://doi.org/10.1016/j.compstruct.2021.115060.
  2. Barjasteh, E.; Nutt, S.R.; Moisture absorption of unidirectional hybrid composites. Composites: Part A 2012, 43, 158-164. https://doi.org/10.1016/j.compositesa.2011.10.003.

Comment 5: In part 2.1, “The CFRP used in this paper was composed of carbon fiber (CF), glass fiber (GF), epoxy resin…”, there is some glass fiber in CFRP. It should not be called CFRP. Please adjust this statement. In addition, the overall volume fraction of fiber is 67%. What are the volume fractions of carbon fiber and glass fiber? 

Response:

Considering the CFRP used in this paper was composed of carbon fiber (CF), glass fiber (GF), epoxy resin, Carbon fiber-reinforced plastic (CFRP) was changed by Carbon/glass fiber reinforced polymer hybrid composite (HFRP). The content of the fiber in the HFRP was tested using a thermal gravimetric analyzer (TGA55, TA Instruments, New Castle, DE, USA) and the content of the fiber was about 67%.

However, the detail information about the volume fractions of carbon fiber and glass fiber is difficult for us to determine because we did not prepare the HFRP by ourselves and obtain the material from a kind of commercial market approaches. On the other hand, this manuscript did not focus on the effect of volume fractions of each fiber on the properties of the molding parts with HFRP, and this issue is probably our next job.

Comment 6: In table 2, CFRP contains 1% ~ 5% PA6. In general, PA6 is a solid state at room temperature, so what is its existing form in CFRP?

Response:

From our observations, PA6 existed as a kind of powder dispersed in phases of other materials.

Comment 7: The micro-grooves had a bottom diameter of 94 μm, a depth of 180 μm and an inclination angle of 12°. Will the CFRP surface be damaged during the preparation of the micro-grooves, such as resin cracking, fiber breaking and interface debonding?

Response:

When the micro-grooves were fabricated on the surface of HFRP, the surface might be damaged during the preparation of the micro-grooves, such as resin cracking, fiber breaking. In order to avoid those problems, we carried out the hot-pressing process perpendicular to the fiber direction to guarantee the direction of the micro-grooves parallel to the fiber direction. The results showed that the above problems could be avoided to the greatest extent.

Comment 8: The tensile strength of composite part is mentioned in the abstract. However, the tensile test method of composite parts is not given in the part 2. Please make relevant supplement. In addition, Table 3 should be placed in part 2.

Response:

More words were added in the revised manuscript to explain the tensile test method of composite parts: “(4) The tensile strength of the composite parts was measured using an universal tensile machine. The tensile force was increased from a preload of 5N and at a tensile speed of 0.01m/s until the joint surface of the composite parts was separated. In order to obtain more reliable experimental results, each tensile curve was obtained from the average data of three same samples.

In addition, based on your useful comment, we placed the Table 3 in part 2.

Comment 9: How many samples are adopted to obtain the stress-strain curves in Fig. 3? Generally, the average value of at least three samples shall be taken.

Response: Data from three samples at the same operating conditions were adopted to determine the stress-strain curves in Fig. 3.

Comment 10: In the last part of the results and discussion, it is suggested that the authors summarize the optimal process parameters to obtain the maximum interface performance.

Response:

More words were added in the Conclusion part to summarize the optimal parameters: “(1) The micro-groove structures effectively increased the injection molding area of the composite parts, which could increase the connection strength between HFRP and PA6. In order to increase the molding properties of the micro-groove structures, the hot-pressing direction should be perpendicular to the fiber direction, which could guarantee the direction of the micro-grooves parallel to the fiber direction.”

Comment 11: The conclusion part should be further enriched according to the content of this paper, including 3-4 small conclusions.

Response:

More words were added into the Conclusion part in the revised manuscript.

Reviewer 2 Report

In this work, micro-grooves were fabricated on the surface of carbon fiber reinforced part (CFRP). The PA6 melt was injected into the mold cavity with the CFRP as an insert. The effects of mold temperature, injection pressure, holding pressure and holding time were investigated on the fracture microstructure and connecting strength of the injection molded parts.

This reviewer has the following comments and criticisms on the manuscript.

1. Overall, this manuscript is not a work of significance. It is an experiment report and lacks originality.

2. Title: “Injection molding CFRP/PA6 composite part by using CFRP with micro-grooves” is not accurate enough.

3. The presentation and language should be largely improved. For example,

(1) Page 2: “On the other hand, the molding process of PA6 includes the injection molding, hot-pressing molding and injection compression molding [14, 15]. Injection molding is a …, which has been widely applied in the molding of polymer parts [16, 17]”. “molding process” is wrong; moreover, Refs [14]~[17] are inappropriate.

(2) Page 2: “As shown in Figure1a, the connecting strength at the interface of CFRP and PA6 had an important influence on the mechanical properties and service life of composite parts”. Obviously, Figure1a does not show such result.

(3) Page 2: “Injection molding was performed with PA6 and CFRP by the injection molding machine … to form composite part”?

(4) Page 3: “2.2. Processing technology”?

(5) Page 4: “…, the paper conducted the injection molding process …”?

……

Author Response

Comment 1: Overall, this manuscript is not a work of significance. It is an experiment report and lacks originality.

Response:

Thanks for your comment. The composite parts prepared in this paper could be used in the automobile interiors and. This fabricating method proposed by this manuscript might provide a way for the automobile industry to find a way to manufacture parts with light quality and high strength.

Comment 2: Title: “Injection molding CFRP/PA6 composite part by using CFRP with micro-grooves” is not accurate enough.

Response:

The title of this manuscript was instead by “An injecting molding method for forming the HFRP/PA6 composite parts”.

Comment 3: The presentation and language should be largely improved. For example,

  • Page 2: “On the other hand, the molding process of PA6 includes the injection molding, hot-pressing molding and injection compression molding [14, 15]. Injection molding is a …, which has been widely applied in the molding of polymer parts [16, 17]”. “molding process” is wrong; moreover, Refs [14]~[17] are inappropriate.

Response:

The “molding process” was instead by “forming method”. In addition, we removed the Ref [17] and cited a new reference [23] in the revising manuscript.

  1. Volpe, V.; Lanzillo, S.; Affinita, G.; Villacci, B.; Macchiarolo, I.; Pantani, R. Lightweight High-Performance Polymer Composite for Automotive Applications. Polymers 2019, 11, 326. https://doi.org/10.3390/polym11020326.
  • Page 2: “As shown in Figure1a, the connecting strength at the interface of CFRP and PA6 had an important influence on the mechanical properties and service life of composite parts”. Obviously, Figure1a does not show such result.

Response:

In order to express more clear, relevant words were corrected. “As shown in Figure1a, HFRP/PA6 composite parts were obtained by injection molding using the HFRP without micro-grooves. Based on previous experimental data, the maximum tensile strength of composite part could reach 4.72 MPa (Figure1b). The connecting strength at the interface of HFRP and PA6 had an important influence on the mechanical properties and service life of composite parts.”

  • Page 2: “Injection molding was performed with PA6 and CFRP by the injection molding injection molding machine … to form composite part”?

Response:

Those words were instead by “The composite part with PA6 and HFRP was formed by the injection molding machine (Babyplast 6/10P, Cronoplast Sl, Spain)”.

  • Page 3: “2.2. Processing technology”?

Response:

“2.2. Processing technology” was instead by “ Steps of experiment”

  • Page 4: “…, the paper conducted the injection molding process …”?

Response:

Those words were instead by “the composite parts with HFRP and PA6 were formed by the injection molding method using………”

Reviewer 3 Report

The manuscript investigates the impact of injection molding parameters on the tensile strength of composite sample made from carbon fiber reinforced composite and polyamide -6. The samples were synthesized by fabricating microgrooves on the carbon fiber reinforced composites and the molten PA-6 during injection molding will embed into those grooves as nails. The ideal parameters were found to be 240 oC for mold temperature, 8 MPa for injection pressure, 8 MPa for holding pressure and 3 s for holding time. I recommend the manuscript to be accepted after the following comments have been taken into consideration.

Comment 1:

Throughout the manuscript the authors mention use glass fiber (GF) whereas in certain description for the sample mentions use of carbon fiber only. Also, CFRP stands for carbon fiber reinforced composite. Example of places where GF is used on the manuscript are given below.

·       Section 2.1

·       Line 67

·       Line 128

·       Line 169

·       Line 172

·       Line 213

·       Line 231

·       Figures 3, 4, ,5 and 6

Comment 2:

The authors don’t mention how the CFRP sample was fabricated. Was it prepared from prepregs or VARTM etc.? Also, it is confusing whether glass fiber and carbon fiber were both used. If glass fiber is used then the author should describe how the two different fibers were stacked. I suggest that the authors revise the process technology section and write in detail how the CFRP was fabricated.

Comment 3:

The authors don’t mention how many samples for each parameter scenario were tested as use of just one sample may not give reliable data. If the authors used one sample for each scenario, I suggest the authors repeat the tests with multiple samples.

Comment 4:

When the mold temperature, injection pressure and hold pressure were varied the authors mention about keeping the holding time constant to 2.5 s which is inconsistent with Table 3.

Comment 5:

There are many typos which need to be corrected. Some of the typos are listed below:

Line 137: “In addition, excessively high mold temperature also caused bubbles generated … …

Line 178: “With the increasing of the injection pressure, the mold filling rate of melt PA6 ... …

The word “effectively” was used twice in the same sentence (Line: 214): “Therefore, the tensile strength between PA6 and CFRP was effectively improved, effectively ensuring … …

Did the authors meant to use hold time instead of hold pressure in Line 233 and 234?

Therefore, the hold pressure … …

Comment 6:

On Figure 5, it is suggested that the authors explain the decrease in the tensile strength when the holding pressure increased from 6 MPa to 7 MPa.

Comment 7:

It is suggested that the authors remove the word “obviously” from line 163.

Author Response

Comment 1: Throughout the manuscript the authors mention use glass fiber (GF) whereas in certain description for the sample mentions use of carbon fiber only. Also, CFRP stands for carbon fiber reinforced composite. Example of places where GF is used on the manuscript are given below.

  • Section 2.1
  • Line 67
  • Line 128
  • Line 169
  • Line 172
  • Line 213
  • Line 231
  • Figures 3, 4, ,5 and 6

Response:

Thanks for your nice comment. We adjusted the statement and replaced " Carbon fiber-reinforced plastic (CFRP)" with " Carbon/glass fiber reinforced polymer hybrid composite (HFRP) ".

Comment2: The authors don’t mention how the CFRP sample was fabricated. Was it prepared from prepregs or VARTM etc.? Also, it is confusing whether glass fiber and carbon fiber were both used. If glass fiber is used then the author should describe how the two different fibers were stacked. I suggest that the authors revise the process technology section and write in detail how the CFRP was fabricated.

Response:

Thanks for your nice comment. The "HFRP" used in our experiments was obtained directly from a kind of commercial market approach. It is difficult for us to determine the detail information like the fiber ratio of the CFRP, but one thing we can confirm is that the glass fiber and carbon fiber were used. In addition, the manufacturing process of the HFRP was not our focus in this article, we will consider this study in the next step.

Comment 3: The authors don’t mention how many samples for each parameter scenario were tested as use of just one sample may not give reliable data. If the authors used one sample for each scenario, I suggest the authors repeat the tests with multiple samples.

Response:

Thanks for your nice comment. Three composite part samples were taken at each parameter scenario for stretching experiments by the universal material testing machine, the tensile curves were obtained by averaging over the measured data. Based on your useful comment, we supplemented the related content in the manuscript.

Comment 4: When the mold temperature, injection pressure and hold pressure were varied the authors mention about keeping the holding time constant to 2.5 s which is inconsistent with Table 3.

Response:

Thanks for your nice comment. We are very sorry for this error we made in the manuscript. When the mold temperature, injection pressure and hold pressure were varied, we indeed kept the holding time constant at 2.5 s. We have corrected “2 s” to “2.5s” in the part of Table 3 within the revised manuscript.

Comment 5: There are many typos which need to be corrected. Some of the typos are listed below:

Line 137: “In addition, excessively high mold temperature also caused bubbles generated … … ”

Line 178: “With the increasing of the injection pressure, the mold filling rate of melt PA6……”

The word “effectively” was used twice in the same sentence (Line: 214): “Therefore, the tensile strength between PA6 and CFRP was effectively improved, effectively ensuring … …”

Did the authors meant to use hold time instead of hold pressure in Line 233 and 234?

“Therefore, the hold pressure … …”

Response:

Thanks for your nice comment. Based on your useful comment, we corrected those typos as follows:

Line 168: “In addition, excessively high mold temperature will also cause the formation of bubbles……”

Line 210: “With the increasing of the injection pressure, the mold filling rate of molten PA6 ... …”

Line 251: We have removed one of the "effectively" with a modification to “Therefore, the tensile strength between PA6 and HFRP was improved, effectively ensuring … …”

Line 270: We are very sorry for this error we made in the manuscript. We have changed “hold pressure” to “hold time” instead.

Comment 6: On Figure 5, it is suggested that the authors explain the decrease in the tensile strength when the holding pressure increased from 6 MPa to 7 MPa.

Response:

Thanks for your nice comment. We added more words in the revised manuscript to explain the reason why the decrease in the tensile strength when the holding pressure increased from 6 MPa to 7 MPa: “When the holding pressure was 6 MPa, the tensile strength for the composite part was reached at 8.29 MPa. As the holding pressure continued to increase, a part of micro-grooves on the surface of HFRP will break, resulting in a small decrease in the tensile strength. However, as the holding pressure increased further, the previously broken part recombined, which caused the tensile strength continued to increase.”

Comment 7: It is suggested that the authors remove the word “obviously” from line 163.

Response:

Thanks for your nice advice. We removed the word “obviously” from line 194.

Reviewer 4 Report

Please find Comments and Suggestions for Authors in the *.pdf file attached.

Main remark is missing reference between obtained results to part geometry.

Author Response

Comment 1: What does CFRP term refer to, carbon fibers or carbon fiber reinforced part? Needed more detailed explanation.

Response:

Thanks for your nice comment. The “CFRP” used in this paper was composed of carbon fiber (CF), glass fiber (GF), epoxy resin, so we adjusted the statement to your recommendation, changing " Carbon fiber-reinforced plastic (CFRP)" to " Carbon/glass fiber reinforced polymer hybrid composite (HFRP) instead. 

Comment 2: 8MPa=80bar -> relatively low injection and packing pressure - lacking information on part geometry.

Response:

Thanks for your nice comment. The sizes of composite parts we finally obtained were small, so the injection pressure and packing pressure required during the experiment were relatively low.

(3) Comment: What are the dimensions of the injected sample?

Response:

Thanks for your nice comment. Based on your useful comment, we made the revision as follows:

Line 134: “The length of the bonding zone of the composite parts was 2 mm, its width was 2 mm, and its height was 2 mm. The overall length of the composite parts was 10 mm.”

(4) Comment: Not universal conclusions nor given process parameters, largely dependent on part geometry. Missing explanation regarding sample part dimensions.

Response:

Thanks for your nice comment. Based on your useful comment, we supplemented the explanation regarding sample part dimensions in the manuscript.

Round 2

Reviewer 1 Report

The authors have responded well to the comments and recommended accepting the current paper.

Author Response

The authors have responded well to the comments and recommended accepting the current paper.

Response:

We are very grateful to you for reviewing the paper so carefully. Your comments are all valuable and very helpful for revising and improving the paper. Thank you for your comments.

Reviewer 3 Report

It could be mentioned in the "Materials" section that the hybrid carbon/glass fiber composite was obtained commercially. This will clarify to the audience that the composite was not fabricated in this study.

Author Response

It could be mentioned in the "Materials" section that the hybrid carbon/glass fiber composite was obtained commercially. This will clarify to the audience that the composite was not fabricated in this study.

Response:

Thanks for your nice comment. More words were added in the "Materials" section: “The HFRP used in this paper was provided by Shenzhen Silver Basis Technology Co., Ltd. It was……”